# A GPX4-dependent cancer cell state underlies the clear-cell morphology and confers sensitivity to ferroptosis

Yilong Zou[1,2], Michael J. Palte[1], Amy A. Deik[1], Haoxin Li[1,2], John K. Eaton [1], Wenyu Wang[1], Yuen-Yi Tseng[1], Rebecca Deasy [1], Maria Kost-Alimova[1], Vlado Dančík[1], Elizaveta S. Leshchiner[1], Vasanthi S. Viswanathan[1], Sabina Signoretti[3], Toni K. Choueiri [4], Jesse S. Boehm [1], Bridget K. Wagner[1], John G. Doench [1], Clary B. Clish [1], Paul A. Clemons [1] & Stuart L. Schreiber [1,2]

Clear-cell carcinomas (CCCs) are a histological group of highly aggressive malignancies commonly originating in the kidney and ovary. CCCs are distinguished by aberrant lipid and glycogen accumulation and are refractory to a broad range of anti-cancer therapies. Here we identify an intrinsic vulnerability to ferroptosis associated with the unique metabolic state in CCCs. This vulnerability transcends lineage and genetic landscape, and can be exploited by inhibiting glutathione peroxidase 4 (GPX4) with small-molecules. Using CRISPR screening and lipidomic profiling, we identify the hypoxia-inducible factor (HIF) pathway as a driver of this vulnerability. In renal CCCs, HIF-2α selectively enriches polyunsaturated lipids, the rate-limiting substrates for lipid peroxidation, by activating the expression of hypoxia-inducible, lipid droplet-associated protein (*HILPDA*). Our study suggests targeting GPX4 as a therapeutic opportunity in CCCs, and highlights that therapeutic approaches can be identified on the basis of cell states manifested by morphological and metabolic features in hard-to-treat cancers.

[1] The Broad Institute, Cambridge, MA 02142, USA. [2] Department of Chemistry and Chemical Biology, Harvard University, Cambridge, MA 02138, USA. [3] Department of Oncologic Pathology, Dana-Farber Cancer Institute and Brigham and Women's Hospital, Harvard Medical School, Boston, MA 02215, USA. [4] Department of Medical Oncology, Dana-Farber Cancer Institute and Brigham and Women's Hospital, Harvard Medical School, Boston, MA 02215, USA. Correspondence and requests for materials should be addressed to S.L.S. (email: stuart_schreiber@harvard.edu)

Clear-cell carcinomas (CCCs), marked by the clear cytoplasm in neoplastic cells during histological analyses, constitute the most frequent and metastatic form of kidney malignancy and the most therapy-resistant form of ovarian carcinomas[1,2]. Though occurring at low incidences, CCCs also arise from a wide range of other tissues, including the cervix, thyroid, liver, and pancreas[3–6]. The broad resistance to current anti-cancer therapies and dismal prognoses in patients with CCCs present a significant unmet medical need.

The clear-cell morphology of CCCs, caused by highly active lipid and glycogen synthesis and deposition, reflects a unique metabolic state of cancer. While these metabolic alterations support cancer progression and promote resistance to immune surveillance and therapies[7,8], directly targeting the primary metabolic liabilities remains challenging due to complications from metabolic plasticity and systemic toxicity of currently available compounds[9,10]. Exploring novel vulnerabilities associated with the metabolic state in CCCs is critical for developing new therapies.

Recently, we generated quantitative sensitivity profiles of 481 "Informer Set" compounds in 887 cancer cell lines from various lineages, including kidney and ovary, and made these data and analysis tools available in the Cancer Therapeutics Response Portal (CTRP)[11–13]. These compounds perturb many distinct nodes in cellular pathways, and therefore are informative for identifying both pan-cancer and tissue-specific vulnerabilities. Here, by interrogating the CTRP, we report that inhibitors of glutathione peroxidase 4 (GPX4), among all nodes of cell circuitry, exhibit the highest selectivity and potency in killing CCC cells. This observation is consistent with a recent report showing that clear-cell renal cell carcinomas (ccRCC), a major subtype of CCCs, are hypersensitive to GPX4 knockdown[14]. Given GPX4's role in selectively detoxifying lipid hydroperoxides, and that GPX4 inhibition triggers ferroptotic cell death (ferroptosis)[15], we postulate that CCC cells are intrinsically susceptible to ferroptosis, and targeting GPX4 represents a therapeutic opportunity in these devastating diseases.

Though the developmental role of ferroptosis has not yet been identified, ferroptotic death is associated with various pathological conditions, including acute kidney injury, hepatocellular degeneration and hemochromatosis, traumatic brain injury, and neurodegeneration[16–20]. Notably, we and others have recently identified a GPX4-dependent state in therapy-resistant cancer cells, including therapy-induced persister cells, that in general resist apoptotic death otherwise induced by the main modalities of cancer treatments—chemotherapy, targeted therapy, and immunotherapy[21–23]. These insights point to ferroptosis-inducing agents as attractive therapeutic strategies for cancer treatment in certain contexts, for example, in minimal residual disease. However, the intrinsic susceptibility of untreated cancers to ferroptosis varies significantly among organ systems and the mechanisms underlying cell type-specific ferroptosis sensitivity are poorly understood.

In the present study, we systematically characterize the mechanisms driving the histotype-specific GPX4 dependency in CCCs. By combining genome-wide CRISPR screening and lipidomic profiling, we highlight the HIF-2α-HILPDA axis as a central driver of this vulnerability. HIF-2α-HILPDA selectively enriches lipids that contain polyunsaturated fatty acyl side chains and induces a ferroptosis-susceptible cell state. Our findings have implications for understanding the mechanisms of the ferroptosis pathway, as well for developing novel treatment options for CCC patients.

## Results

### Sensitivity to GPX4 inhibition-induced ferroptosis in CCCs.
To search for druggable vulnerabilities in CCCs, we systematically interrogated the CTRP datasets [portals.broadinstitute.org/ctrp/] to identify CCC-selective chemicals[12,13]. The CCC cell-line collection in CTRP comprises 17 clear-cell renal cell carcinoma (ccRCC) and 9 ovarian CCC (OCCC) cell lines[24]. In contrast to the low efficacy of conventional chemotherapies such as paclitaxel, three GPX4 inhibitors emerged as the most potent and selective compounds for killing CCC cells: (1S, 3R)-RSL3 (RSL3), ML210 and, ML162[15,21] (Fig. 1a, b, Supplementary Fig. 1a). The GPX4 inhibitor sensitivity in CCC cells was stronger than the sensitivity of any specific solid tumor lineage (Fig. 1c, Supplementary Fig. 1b). Moreover, the chemical sensitivity was confirmed by strong genetic dependence on GPX4 using both CRISPR and shRNAs in the Cancer Dependency Map (DepMap) database[25], which explores genetic dependencies (Supplementary Fig. 1c). GPX4 uses glutathione to detoxify lipid hydroperoxides selectively and acts as a gatekeeper for ferroptosis, an iron-dependent cell-death pathway[15]. Our results imply that CCCs are intrinsically vulnerable to ferroptosis.

We first validated the GPX4 dependency in several frequently used ccRCC cell lines, including 786-O, 769-P, OS-RC2, and RCC10RGB via small-molecule, CRISPR or shRNA-mediated GPX4 inhibition (Fig. 1d, Supplementary Fig. 2a–c). GPX4 inhibition-induced cell death in ccRCC cells was completely blocked by treatment with ferroptosis rescue agents ferrostatin-1 (Fer-1) or liproxstatin-1 (Lip-1) (Fig. 1e). Results from our characterizations using covalent GPX4 inhibitors were consistent with previous studies using erastin and L-buthionine-S,R-sulfoximine (BSO), compounds that target distinct steps in the creation of glutathione in cells, in ccRCC cells[14]. Moreover, ML210-treatment induced rapid accumulation of lipid radicals in ccRCC but not BFTC909 cells, as reported by BODIPY-C11, confirming the involvement of ferroptotic cell death (Fig. 1f, Supplementary Fig. 1d). This vulnerability to ferroptosis was recapitulated in 786-O xenografts in vivo (Fig. 1g–i), and in patient-derived, primary ccRCC cell lines (Fig. 1j, k, Supplementary Table 1). Notably, ccRCC cells exhibited substantially higher sensitivity to ferroptosis than normal renal cells (Fig. 1d, k), which possess basal level of ferroptosis sensitivity[26,27], indicating the presence of a therapeutic window for inducing ferroptosis as a ccRCC treatment strategy.

In ovarian cancers, OCCC cells exhibited significantly higher sensitivity to GPX4 inhibitors and lower sensitivity to paclitaxel than other ovarian carcinoma lines in average in CTRP (Supplementary Fig. 2d). The ferroptosis susceptibility was strong in OCCC cell lines ES-2, OVISE, and TOV21G, but weak in at least one high-grade serous carcinoma (HGSC) cell line OV-90 (Fig. 1l, Supplementary Fig. 2e). Lip-1 treatment rescued OCCC cells from ML210 or RSL3-induced cell death (Supplementary Fig. 2e). Moreover, CRISPR or shRNA-mediated GPX4-depletion significantly reduced the viability of ES-2 cells (Supplementary Fig. 2a–c). The shared ferroptosis sensitivity between ccRCC and OCCC is consistent with their resemblance at the transcriptome level[28,29]. Additionally, mRNA levels of frequently used CCC markers HNF-1β, PAX8, PLIN2, and PLIN3[30–33] strongly correlate with sensitivity to GPX4 inhibitors in CTRP (Supplementary Fig. 2f). Collectively, these results indicate that CCC cells are intrinsically susceptible to GPX4 inhibition-induced ferroptosis.

### HIF-1/2α mediates sensitivity to ferroptosis in CCCs.
Although ferroptosis is frequently triggered under pathological conditions such as ischemia/reperfusion and traumatic brain injuries[34], how intrinsic ferroptosis sensitivity arises is poorly understood. Illuminating the mechanisms underlying ferroptosis susceptibility is important for identifying the right patient cohort that would benefit from ferroptosis-inducing agents. Notably, the highly

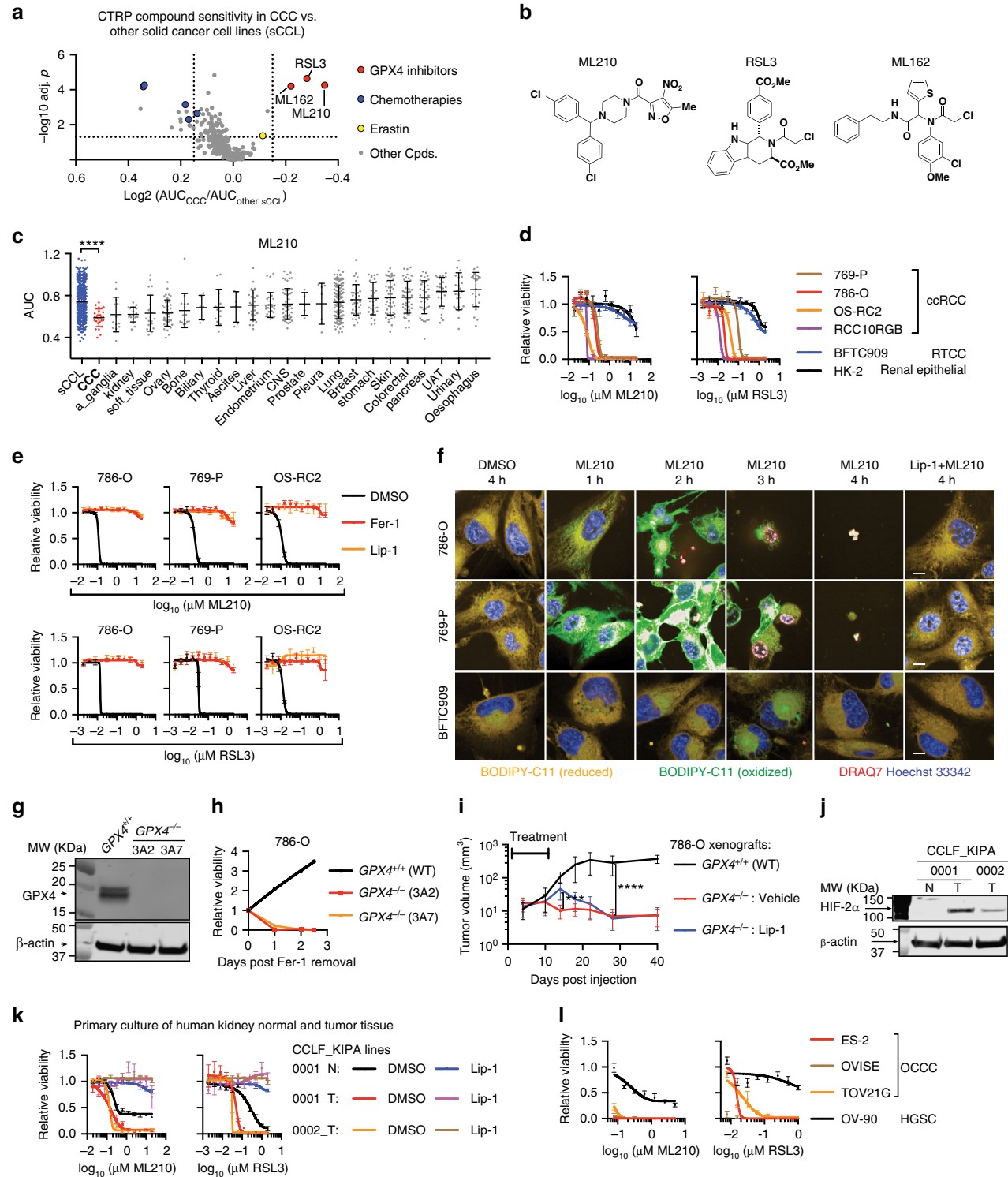

variable sensitivity to ferroptosis across CTRP models appears independent of the relative levels of *GPX4* mRNA (Supplementary Fig. 3a). While CCCs arising from different lineages remain genetically distinct, we focused on characterizing ccRCCs, the most frequent and genetically defined CCC subtype, by performing a genome-wide CRISPR suppressor/resistance screen in 786-O cells to identify mediators of ML210 sensitivity (Fig. 2a, Supplementary Data 1–3). Among the genes required for ML210 sensitivity in all three time-points, the top hits included

acyl-CoA synthetase long-chain family member 4 (*ACSL4*) and *Kelch*-like ECH associated protein 1 (*KEAP1*), two genes that were previously implicated in ferroptosis[35,36] (Fig. 2b); we verified both genes as required for ferroptosis in CCCs (Supplementary Fig. 3b–i). Importantly, we found that genes related to the HIF pathway, including *EPAS1* (encoding HIF-2α), *EP300*, *FOSL1*, *CITED1*, as well as *ARNT* (encoding HIF-1β) are enriched in the top screening hits in one or multiple conditions[37,38] (Fig. 2b). HIF-2α is a driver of ccRCC oncogenesis and

**Fig. 1** Clear-cell carcinoma cells are intrinsically sensitive to GPX4 inhibition-induced ferroptosis. **a** Volcano-plot showing compound sensitivity comparison by normalized area-under-curve (AUC) values between clear-cell carcinoma (CCC) cells ($n = 26$) and other solid tumor cancer cell lines (sCCL) ($n = 634$) in CTRP. Cpds, compounds. **b** Chemical structures of GPX4 inhibitors ML210, RSL3, and ML162. **c** Scatterplot of AUCs for ML210 in sCCL (blue), CCC (red) or cancer cell lines from each tissue. Tissue types are ordered by the average AUC values. Abbreviations: CNS, central nervous system; UAT, upper aerodigestive tract; a_ganglia, autonomic ganglia. Mann–Whitney–Wilcoxon test, ****$p < 0.0001$. **d** Viability curves for the indicated cells treated with ML210 or RSL. $n = 4$. Representative plot of experiments repeated three times. **e** Viability curves for the indicated cells treated with ML210 or RSL3 plus indicated DMSO, liproxstatin-1 (Lip-1) or ferrostatin-1 (Fer-1). Representative plot of experiments repeated three times. **f** Fluorescent images of BODIPY-C11 stained 786-O, 769-P, and BFTC909 cells treated with ML210 plus DMSO or Lip-1 for the indicated time periods. Scale bars: 10 μm. **g** Immunoblot showing GPX4 protein levels in $GPX4^{+/+}$ 786-O or $GPX4^{-/-}$ clones 3A2 and 3A7. **h** Viability curves for WT 786-O, 3A2, and 3A7 over a 2.5-day time course after Fer-1 removal. Representative plot of experiments repeated three times. **i** Tumor volume measurements of subcutaneous xenografts of WT 786-O or $GPX4^{-/-}$ 3A2 cells ($n = 10$ mice, two tumors per mouse). $GPX4^{-/-}$ tumor-bearing mice were divided to a Lip-1 treated group and a vehicle-treated group. Treatment lasted for the first 10 days. Two-tailed $t$-test, ***$p < 0.001$, ****$p < 0.0001$. **j** Immunoblot showing HIF-2α protein levels in primary human ccRCC cell lines CCLF_KIPA_0001_T, CCLF_KIPA_0002_T, and normal renal-cell culture CCLF_KIPA_0001_N matched with CCLF_KIPA_0001_T. **k** Viability curves for indicated cell lines treated with ML210 or RSL3 plus DMSO or 0.5 μM of Lip-1. $n = 4$. Representative plot of experiments repeated twice. **l** Viability curves for ovarian clear-cell carcinoma (OCCC) cell lines ES-2, OVISE, and TOV21G, and high-grade serous carcinoma (HGSC) line OV-90 treated with ML210 (left) or RSL3 (right). $n = 4$. Representative plot of experiments repeated twice. Error bars: ±s.d. β-Actin was used as loading controls for immunoblots

acquisition of the clear-cell morphology[39,40], and its emergence as a ferroptosis regulator is consistent with a prior study revealing that VHL-restoration diminished the sensitivity to erastin and BSO in RCC4, another ccRCC cell line[14]. Gene suppression with independent sgRNA and shRNA libraries validated this pathway as mediators of ML210 sensitivity in 786-O cells (Fig. 2c, Supplementary Data 9 and 10). HIF-2α-dependent sensitivity to ferroptosis was also observed in ccRCC cells expressing individual HIF-2α-targeting sgRNAs and shRNAs, in single-cell $EPAS1^{-/-}$ clones with or without restored $EPAS1$-GFP expression, as well as HIF-2α/GPX4 double knockouts (Fig. 2d–h, Supplementary Fig. 4a–f and Supplementary Data 8). While loss of HIF-2α did not compromise the proliferation rate of ccRCC cells in vitro[41,42] (Supplementary Fig. 4g), HIF-2α ablation significantly reduced lipid peroxidation levels (Supplementary Fig. 4h–i), providing a strong indication of reduced susceptibility to ferroptosis.

The HIF-2α-induced ferroptosis sensitivity underscores a prominent example of oncogene-induced vulnerability in ccRCC. Notably, cancer cells with $VHL$ mutations exhibited greater dependence on GPX4 than $VHL$ wildtype cells in a pan-cancer DepMap analysis (Supplementary Fig. 4j). Intriguingly, OCCC tumors mimic the hypoxia response in the endometrium cyst microenvironment with activated HIF-1α[43]. Notably, HIF-1α-depletion by CRISPR diminished the sensitivity to ferroptosis in ES-2 cells (Supplementary Fig. 4k–l). Collectively, our results indicate that the HIF pathway is a central driver of ferroptosis susceptibility in CCCs. In addition, HIF prolyl hydrolase 2 ($EGLN1$) diminishes ferroptosis susceptibility by destabilizing HIF-1α in A549 non-small cell lung cancer cells[44], implying a more general role of this pathway in ferroptosis in other cancer contexts.

**HIF-2α selectively enriches polyunsaturated lipids in CCCs.** Ferroptosis is executed by peroxidized membrane phospholipids, particularly phosphatidylethanolamines (PEs) that contain poly-unsaturated fatty acyl (PUFA) chains including arachidonic acid (C20:4) and docosahexaenoic acid (C22:6)[45,46]. Although HIFs drive extensive reprogramming of lipid metabolism and promote lipid storage in cancer cells, how these metabolic alterations are associated with ferroptosis sensitivity is unknown. To determine how the HIF pathway drives ferroptosis sensitivity, we performed lipidomic profiling in $EPAS1^{-/-}$ single-cell 786-O clones (Supplementary Fig. 5a, b, Supplementary Data 4). HIF-2α-depletion induced a profound shift in the lipidome of 786-O cells, with significant loss in triacylglycerols (TAGs), the major components of lipid droplets, and in phospholipids (Fig. 3a, Supplementary

Fig. 5c, d). Remarkably, PUFA-TAGs exhibited the most significant reduction in response to HIF-2α-depletion compared with TAGs containing saturated/monounsaturated fatty acyl chains (SFA/MUFA-TAGs) (Fig. 3a, b). Alterations in TAG saturation levels during hypoxia response was also previously noted in ccRCC cells[47]. Moreover, most PEs and PE-plasmalogens (ePEs), including the ferroptosis-relevant C36:4, C38:4/5/6 and C40:6 PEs and C36:5, C38:5 and C40:7 ePEs, were significantly reduced in $EPAS1^{-/-}$ cells (Fig. 3c, d). Finally, free PUFA levels were also strongly dependent on HIF-2α activity, whereas free SFA/MUFAs were less affected by HIF-2α status (Fig. 3e, Supplementary Data 5). Most of these alterations were reverted by HIF-2α-GFP overexpression, supporting these events as specifically driven by HIF-2α (Fig. 3a–e, Supplementary Fig. 5a–d). Importantly, exogenous PUFA (arachidonic acid, C20:4) treatment significantly sensitized WT or HIF-2α-depleted 786-O and 769-P cells to ferroptosis (Fig. 3f). Taken together, these data suggest that HIF-2α drives ferroptosis susceptibility by selective enrichment and incorporation of PUFAs into glycerolipids (TAGs and phospholipids).

By interrogating a previous lipidomics dataset comprising 49 ccRCC normal/tumor tissue pairs[48], we found that human ccRCC tumors exhibited higher levels of PUFA-PE/ePEs and PUFA-PC/ePCs than normal renal tissues (Fig. 3g). These PUFA-lipids were further enriched in high-grade tumors (stage III/IV) when compared with low-grade samples (stage I/II) (Supplementary Fig. 5e). Thus, human ccRCC tumors, which commonly possess constitutively active HIF-2α[48], are in a PUFA-lipid-enriched cell state and are likely to be sensitive to ferroptosis.

**HILPDA mediates HIF-2α's ferroptosis sensitization activity.** To determine the downstream mediators of HIF-2α's activity in stimulating the selective enrichment of PUFA-lipids and driving ferroptosis sensitivity, we first identified HIF-2α-dependent genes and then re-expressed each gene in $EPAS1^{-/-}$ cells to identify candidate genes that restore sensitivity to ferroptosis (Fig. 4a). We were able to collect cDNAs for 77 of the 149 HIF-2α-activated genes identified by RNA-Seq, including 9 of the 11 lipid metabolism genes (Fig. 4b, Supplementary Data 6). RNA-Seq and western blot analyses also ruled out any significant changes in ACSL family expression induced by HIF-2α (Supplementary Fig. 6a, b). With $ALOX15$ (15-Lipoxygenase-1) as a positive control, we identified hypoxia-inducible, lipid droplet-associated protein ($HILPDA$, also known as HIG2) and G0/G1 Switch 2 ($G0S2$) as top re-sensitization factors (Fig. 4c, d, Supplementary Fig. 6c, d). HILPDA and G0S2 share a homologous

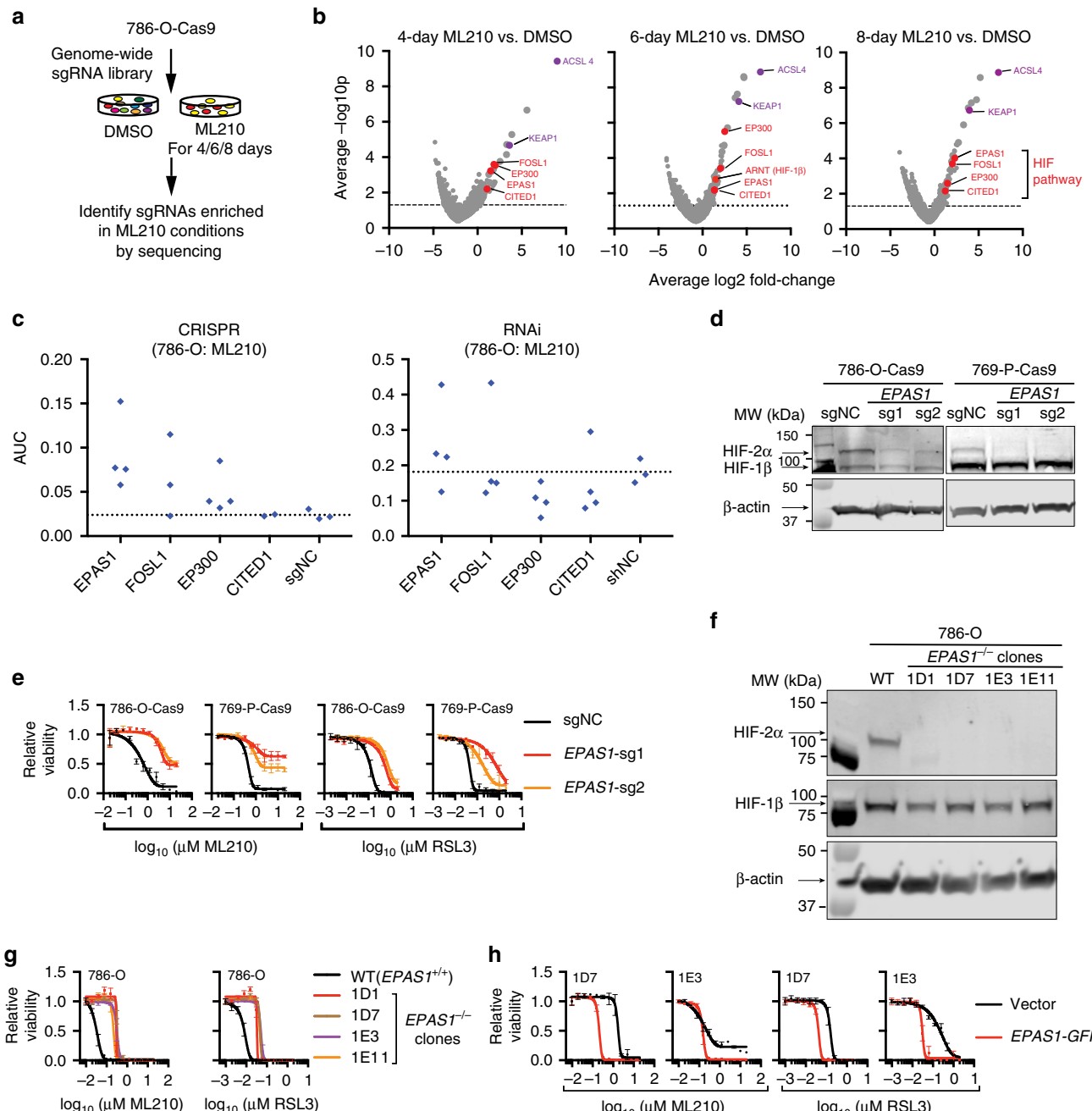

**Fig. 2** Genome-wide CRISPR screen identifies HIF-2α as a driver of ferroptosis susceptibility. **a** Experimental scheme describing the genome-wide CRISPR resistance screening to identify mediators of ML210 sensitivity in 786-O cells. **b** Volcano plot highlighting top enriched CRISPR hits in 786-O cells treated with ML210 for 4, 6 or 8 days. Red genes, HIF pathway genes. Purple genes, representative known ferroptosis regulators. **c** Relative AUC values of the Cas9/sgRNA (CRISPR) or shRNA (RNAi) transfected 786-O cells treated with a 7-point, 2-fold dilution series of ML210. The viability of cells expressing each sgRNA/shRNA (blue dots) was normalized to the respective DMSO-treated condition. AUC values were normalized to 1 as the total area-under-curve for the concentration range of ML210. **d** Immunoblot showing the HIF-2α/HIF-1β protein levels in control (sgNC) or *EPAS1*-targeting sgRNA-expressing 786-O-Cas9 and 769-P-Cas9 cells. **e** Viability curves of control (sgNC) or *EPAS1*-targeting sgRNA-expressing 786-O-Cas9 and 769-P-Cas9 cells treated with indicated concentrations of ML210 or RSL3. Representative plot of experiments repeated three times. **f** Immunoblot showing HIF-2α and HIF-1β protein levels in wildtype (WT, *EPAS1*+/+) 786-O cells and four *EPAS1*−/− single-cell clones generated by CRISPR/Cas9. **g** Viability curves for WT *EPAS1*+/+ 786-O or *EPAS1*−/− clones treated with indicated concentrations of ML210 or RSL3. Representative plot of experiments repeated three times. **h** Viability curves for *EPAS1*−/− 786-O single-cell clones 1D7 and 1E3 expressing vector or *EPAS1-GFP*, then treated with indicated concentrations of ML210 or RSL3. Representative plot of experiments repeated three times. β-Actin was used as loading controls in immunoblots

PNPLA-binding motif (Supplementary Fig. 6e), and each can act as a co-inhibitor of adipose triglyceride lipase (ATGL, encoded by *PNPLA2*)[49,50], the rate-limiting enzyme for TAG hydrolysis. In contrast, overexpressing another HIF-2α-regulated, lipid droplet-associated protein perilipin2 (*PLIN2*) did not alter GPX4 inhibitor sensitivity[40] (Fig. 4c, Supplementary Fig. 6c), suggesting that specific lipid remodeling activity downstream of HILPDA/G0S2 is required for ferroptosis sensitization.

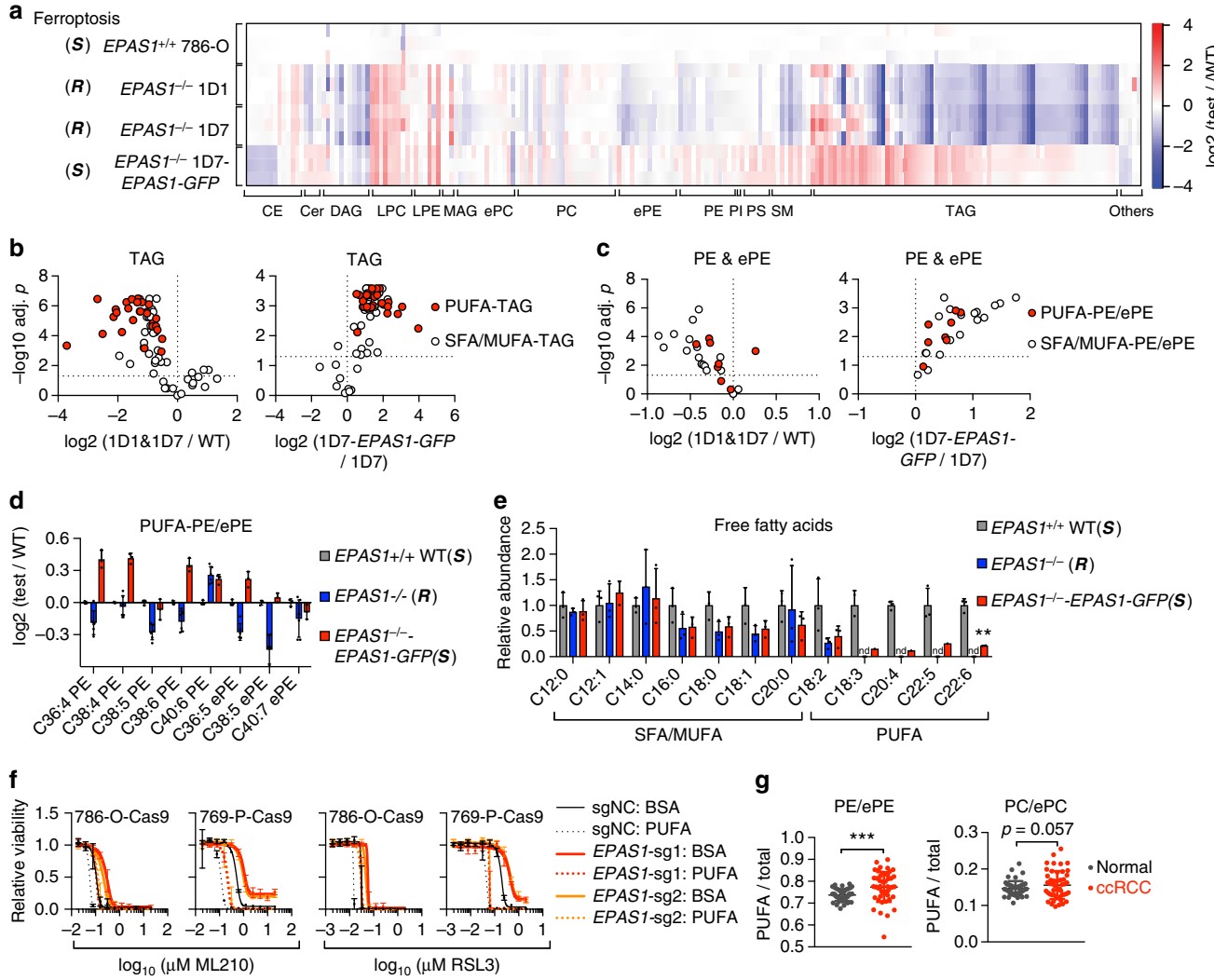

**Fig. 3** HIF-2α selectively enriches polyunsaturated lipids. **a** Heatmap representing the relative lipid abundances in indicated cell lines. The abundance of each lipid species is normalized to the mean of that in the *EPAS1*[+/+] 786-O WT cells and the ratios are log2 transformed. The lipids are grouped by classes, and within each class, the lipid species are ordered first with increasing carbon number, then with increasing unsaturation levels. Abbreviations: CE, cholesterol ester; Cer, ceramide; MAG, monoacylglycerol; DAG, diacylglycerol; TAG, triacylglycerol; LPC, lysophosphatidylcholine; LPE, lysophosphatidylethanolamine; PC, phosphatidylcholine; PE, phosphatidylethanolamine; ePC, (vinyl ether-linked) PC-plasmalogen; ePE, (vinyl ether-linked) PE-plasmalogen; PI, phosphatidylinositol; PS, phosphatidylserine; SM, sphingomyelin. Blue: down-regulated relative to the WT cells, red: upregulated relative to the WT cells. The wave-like pattern in the TAG class corresponds to the more significant losses in the polyunsaturated fatty acyl (PUFA)-TAGs than saturated/monounsaturated fatty acyl (SFA/MUFA)-TAGs in response to HIF-2α-depletion. *S*, ferroptosis-sensitive; (*R*), ferroptosis-resistant. **b** Volcano plots showing the changes in TAGs grouped as PUFA-TAGs (red fill) and SFA/MUFA-TAGs (white fill) between the indicated cell lines. $n = 3$, two-tailed t-test. **c** Volcano plots showing changes in PE and ePE lipids grouped as PUFA-PE/ePEs (red fill) and SFA/MUFA-PE/ePEs (white fill) between the indicated cell lines. **d** Bar graph representing the relative abundances of the indicated PUFA-PE/ePE lipids in the labeled groups. Log2 fold changes relative to 786-O WT cells are presented for each condition. $n = 3$, error bars: ±s.d. **e** Bar graph representing the relative abundances of the indicated free fatty acids grouped as PUFAs or SFA/MUFAs in the indicated conditions. nd, not-detectable under the experimental condition. $n = 3$, error bars: ±s.d. **f** Viability curves of 786-O-Cas9 and 769-P-Cas9 cells expressing control (sgNC) or *EPAS1*-targeting sgRNAs, first treated with BSA or PUFA (arachidonic acid, C20:4) for 3 days, then treated with indicated concentrations of ML210 or RSL3 for 48 h. Representative plot of experiments repeated twice. **g** The ratios between PUFA-PE/ePE and total PE/ePE (left), and between PUFA-PC/ePC and total PC/ePCs in ccRCC tumor samples ($n = 49$; red) and the matched normal tissues ($n = 49$; gray) from previously reported lipidomics datasets. Student's T-test, ***$p < 0.001$

While qRT-PCR confirmed the HIF-2α-dependent expression of HILPDA and G0S2 (Supplementary Fig. 6f), analyses of previous ChIP-Seq data in 786-O cells[51] revealed HIF-2α/HIF-1β binding to genomic loci adjacent to *PLIN2* and *HILPDA* but not to *G0S2* (Supplementary Fig. 6g), supporting *PLIN2* and *HILPDA* as direct HIF-2α target genes; whereas the regulatory mechanisms of HIF-2α on *G0S2* expression remain to be characterized. shRNA-mediated knockdown of endogenous *HILPDA* diminished GPX4 inhibitor sensitivity in 786-O cells (Fig. 4e–g).

However, detecting endogenous G0S2 protein expression in CCC cells proved challenging even with multiple commercially available and validated antibodies; and four sequence-independent *G0S2*-targeting shRNAs did not alter the ferroptosis sensitivity in 786-O cells (Supplementary Fig. 6h, i). Additionally, 786-O-Cas9 cells expressing both a *HILPDA*-targeting shRNA and a high-score *G0S2*-targeting sgRNA nearly phenocopied the ferroptosis sensitivity of cells expressing only the *HILPDA*-shRNA (Supplementary Fig. 6j, k). Altogether, these results

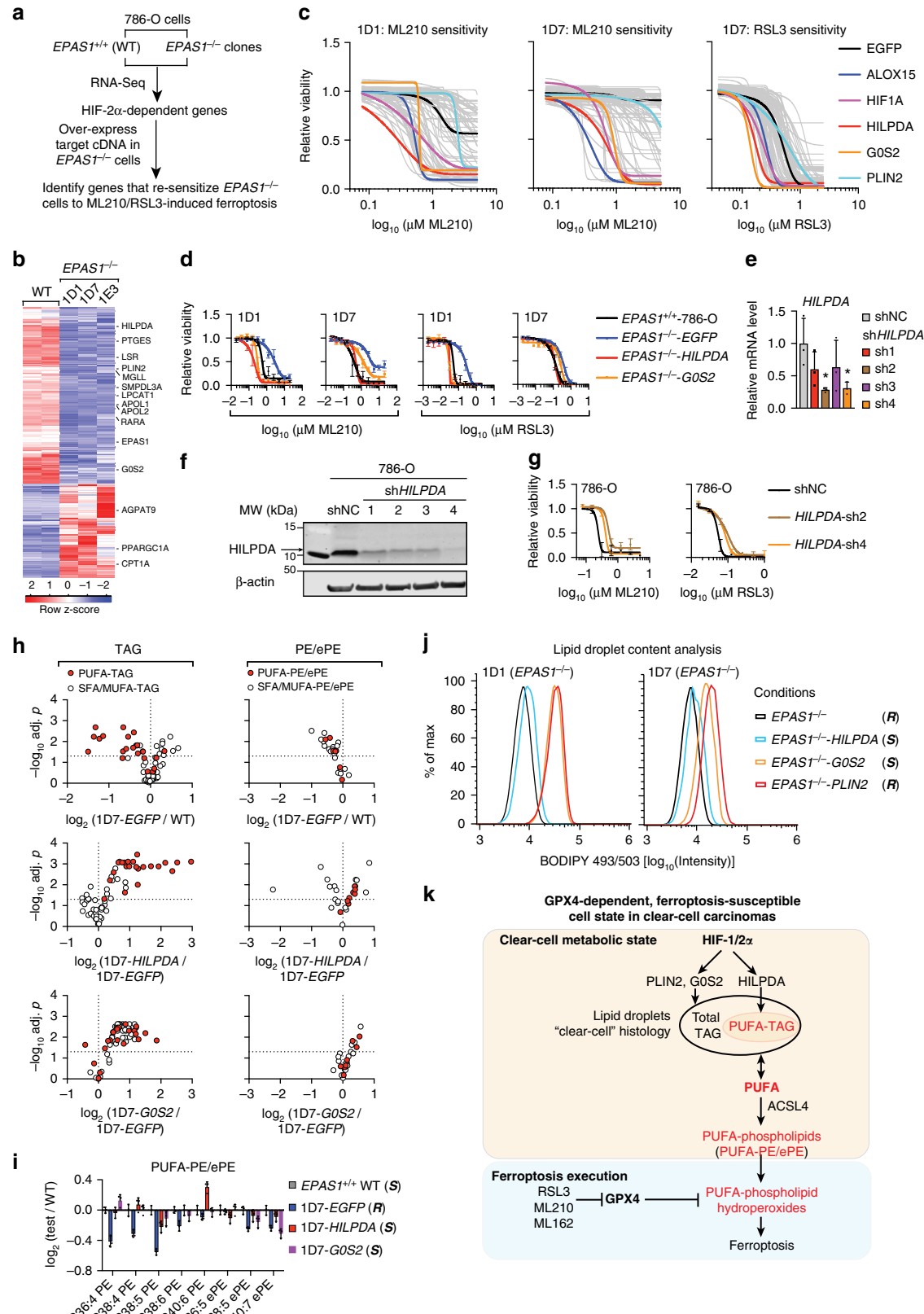

suggest that HILPDA is necessary and sufficient to mediate HIF-2α's activity in driving ferroptosis susceptibility. While over-expressed G0S2 is sufficient to drive ferroptosis sensitivity, the role of endogenously expressed G0S2 protein in ferroptosis is likely insignificant.

In the cDNA screening experiment, we also included HIF-1α in the cDNA library and found that HIF-1α expression re-sensitized HIF-2α-null cells to ferroptosis (Fig. 4c). Importantly, *HILPDA* is also a HIF-1α target gene, and is up-regulated in OCCCs compared with other ovarian carcinomas[52,53]. These observations

**Fig. 4** HILPDA enriches polyunsaturated lipids and promotes ferroptosis sensitivity downstream of HIF-2α. **a** Scheme summarizing the experimental strategy for identifying the HIF-2α target genes mediating ferroptosis susceptibility in 786-O cells. **b** Heatmap showing the RNA-Seq analysis of WT 786-O cells and three *EPAS1*[−/−] clones (1D1, 1D7, and 1E3). HIF-2α-dependent known lipid metabolism genes are highlighted. **c** Viability curves of cDNA screening results in *EPAS1*[−/−] clones treated with a 7-point, 2-fold dilution series of ML210 or RSL3. Viability is relative to that of each cell line under zero ML210 or RSL3 treatment, respectively. $n = 4$, error bars are omitted for visual clarity. **d** Viability curves of HILPDA, G0S2, or EGFP-overexpressing *EPAS1*[−/−] 1D1 and 1D7 cells treated with ML210 or RSL3 for 48 h. Viability is relative to the respective DMSO-treated conditions. $n = 4$, Representative plot of experiments repeated three times. **e** qRT-PCR analysis of *HILPDA* mRNA levels in 786-O cells expressing shNC or sh*HILPDA*s. B2M was used as an internal control. $n = 3$. Student's *T*-test. *$p < 0.05$. **f** Immunoblot analysis of HILPDA protein levels in 786-O cells expressing shNC or sh*HILPDA*s. β-Actin was used as a loading control. **g** Viability curves for 786-O cells expressing shNC or two most effective sh*HILPDA*s under indicated concentrations of ML210 or RSL3. Viability is relative to the DMSO-treated conditions, $n = 4$. Representative plot of experiments repeated three times. **h** Volcano plots showing the changes in each TAG (left panels) and PE/ePE (right panels) species grouped as PUFA-containing (red fill) or SFA/MUFA-only (white fill) lipids between the indicated cell lines. $n = 3$. **i** Bar graph showing the relative abundances of the PUFA-PE/ePEs in the conditions tested. $n = 3$. **S**, ferroptosis-sensitive; (***R***), ferroptosis-resistant. **j** Lipid droplet abundances analyzed by flow cytometry quantitation of BODIPY-493/503 signal in 1D1 and 1D7 *EPAS1*[−/−] cells expressing exogenous EGFP, HILPDA, G0S2 or PLIN2. Representative plot of experiments repeated three times. **k** Scheme summarizing the molecular network driving the intrinsic GPX4 dependency and ferroptosis susceptibility in clear-cell carcinomas. Abbreviations: PUFA, polyunsaturated fatty acids, e.g. arachidonic acid (C20:4); TAG, triacylglycerols; PE, phosphatidylethanolamine; ePE, vinyl ether-linked PE-plasmalogens. Metabolites highlighted in red indicate promoters of ferroptosis susceptibility. Error bars: ±s.d

resonate with the activity of HIF-1α in inducing ferroptosis sensitivity in OCCC cells (cf. Supplementary Fig. 4k, l), and further substantiate the key role of the HIF pathway in ferroptosis susceptibility in CCCs.

**HILPDA enriches polyunsaturated lipids downstream of HIF-2α.** We next characterized the mechanisms by which HILPDA mediates ferroptosis susceptibility in CCCs through lipidomic profiling. Remarkably, HILPDA expression in *EPAS1*[−/−] cells selectively restored the levels of most PUFA-PE/ePEs and PUFA-TAGs, but barely impacted SFA/MUFA-lipids (Fig. 4h, i, Supplementary Fig. 7a–f, Supplementary Data 7). On the other hand, G0S2-overexpression induced a global up-regulation of phospholipids and TAGs regardless of saturation levels (Fig. 4h, i, Supplementary Fig. 7a–f). These results suggest that although HILPDA and G0S2 share similar functional properties[49,50], they exhibit significantly different selectivity in modulating the lipidome, potentially through engaging distinct target proteins. While elevated PUFA-lipid levels are key risk factors for high ferroptosis sensitivity[46], our results indicate that HIF-2α's ferroptosis sensitization role is mediated primarily by HILPDA.

The profound impact of HILPDA on TAG abundance prompted us to assess whether HILPDA also drives the clear-cell phenotype. It was previously revealed that HIF-2α actively promotes lipid deposition and storage in lipid droplets by activating PLIN2 in ccRCC cells[40]. By quantitating lipid droplet (LD) contents, we found that HILPDA induced a modest increase, while G0S2 and PLIN2 induced a strong increase in LD abundances (Fig. 4j, Supplementary Fig. 7g). These results are consistent with PUFA-TAGs, i.e., the HILPDA-dependent species, being a small proportion of total TAGs in ccRCC cells. Collectively, these data suggest that HILPDA is a participant, though not a dominant factor in the HIF-2α-regulated molecular program that stimulates the clear-cell morphology (Fig. 4k).

## Discussion
Patients with CCCs often face systemic resistance, high rates of metastasis and poor prognosis. In this study, we systematically characterized the dependencies of CCCs, and applied high-throughput screening, functional genomics, and metabolomics to dissect the underlying mechanisms. We found that CCCs from distinct lineages share an intrinsic vulnerability to GPX4 inhibition-induced ferroptosis, identified the HIF-HILPDA pathway as the key molecular modality that links this sensitivity with the unique clear-cell metabolic state and morphology, and suggest

GPX4 as a therapeutic target in CCCs (Fig. 4k). Our work points toward the significant potential in combining conventional histology with modern chemical and genetic profiling to reveal important biological and therapeutic insights in difficult-to-treat cancers.

Our study highlights an important role of the HIF pathway in driving ferroptosis sensitivity in cancers, and implies that a GPX4-dependent cell state may be shared by other HIF-α-active cancers, for example, pheochromocytoma and paraganglioma (PCPG). PCPGs are two rare, hard-to-treat neuroendocrine malignancies that display frequent mutations in the VHL/HIF pathway[54]. Importantly, PCPG tumors also exhibit aberrant cytoplasmic lipid accumulation[54], hence testing their GPX4 dependency merits future investigation. Furthermore, as the HIF pathway drives pleiotropic downstream events in cancers, it will be intriguing to examine whether tumors that acquire resistance to the recently developed HIF-2α inhibitors[55,56] retain their ferroptosis susceptibility and sensitivity to GPX4 inhibition, and if so, whether GPX4 inhibitors will be useful to treat these otherwise drug-resistant cancers.

Our work also provides insights into the mechanisms underlying ferroptosis. First, while lipid droplets (LDs) in CCCs have been considered labile organelles that protect cells from lipotoxicity[40], our data imply that PUFA-TAGs in LDs of CCCs contribute to ferroptosis susceptibility. Of note, our PLIN2-overexpressing *EPAS1*[−/−] 786-O cells demonstrated that aberrant LD accumulation is not sufficient to drive ferroptosis susceptibility (Fig. 4c, j). Instead, our lipidomic profiling and functional analyses suggest that the unsaturation level of LD lipids plays a more direct role in dictating sensitivity to ferroptosis, potentially by acting as reservoirs of PUFAs in non-toxic forms and sources for PUFA-phospholipid synthesis. Though the physiological role of PUFA-TAGs/phospholipids in cancers remain unclear, cancer stem cells in several contexts, including ovarian and breast cancers, exhibit increased lipid unsaturation levels[11,57], pointing toward broader applications of ferroptosis-inducing agents in conquering cancer stemness and metastasis.

Secondly, our study emphasizes the potent activity of HILPDA in inducing a ferroptosis-susceptible state downstream of HIF-2α in CCC cells. This activity is mediated through HILPDA's previously overlooked selectivity toward enriching PUFA-TAGs/phospholipids over SFA/MUFA-lipids. Although HILPDA and G0S2 were both shown to repress ATGL activity[49,58], the distinct lipidomic profiles in *HILPDA*-, and *G0S2*-overexpressing cells characterized here strongly suggest that an ATGL-independent activity of HILPDA is present. This activity, coupled with the

remarkable selectivity toward PUFA-lipids, is likely contributing to ferroptosis susceptibility in CCC and other contexts. The molecular basis of HILPDA's lipid selectivity and the mechanisms mediating the interconversion between PUFA-TAGs and PUFA-phospholipids merit future studies. Recently, a mutant allele of *PNPLA3*, the closest homolog of ATGL (*PNPLA2*), was reported to convert selectively PUFA-TAGs to PUFA-phospholipids in non-alcoholic fatty liver disease[59]. Though *PNPLA3* mRNA levels appear below the detection limit in CCC cells in DepMap, similar biochemical pathways may be involved in converting TAGs to phospholipids. Nonetheless, the insights provided in this study shed light on the highly complex and dynamic nature of LDs[60], and highlight how alterations in the lipidome can result in different cell states. Additionally, our results imply that HILPDA expression may be used as a biomarker to predict sensitivity to GPX4-targeting agents in patients.

While most current apoptosis-inducing treatment modalities for clear-cell tumors are challenged by complications from low response rates and emergences of resistance, the unique substrate specificity of GPX4 and its lack of functional redundancy from other peroxidases offer an appealing paradigm in the way that these cancers may be targeted. However, due to the poor bioavailability of current small-molecule GPX4 inhibitors, the in vivo efficacy of chemical inhibition of GPX4 in cancer models remains to be demonstrated. While a recently reported GPX4-targeting strategy suggests that it may be possible to overcome the liabilities associated with conventional GPX4 inhibitors[61], developing novel GPX4 inhibitors with improved pharmacokinetics and pharmacodynamics profile warrants further investigation.

In summary, we identify a histotype-specific ferroptosis susceptibility and GPX4-dependency in CCCs, delineate the genetic and metabolic basis for this dependency, and illuminate the principles of ferroptotic cell death. These insights are potentially translatable toward novel therapies for CCCs and other diseases involving ferroptosis.

## Methods

**Cell lines and culture conditions**. 786-O, 769-P, OS-RC2, ES-2, and OVISE cells were cultured in RPMI-1640 (Gibco) media; RCC10RGB, BFTC909, and HEK-293T cells were cultured in DMEM (Gibco) media; TOV21G and OV-90 cells were cultured in MCDB 105:Medium 199 (1:1, Gibco) mixed media. All media were supplemented with 10% fetal bovine serum (Gibco) and 1% penicillin/streptomycin. Medium for OV-90 cells was supplemented with 1.5 g/L of sodium bicarbonate. HK-2 cells were cultured in keratinocyte serum-free medium (Gibco) supplemented with 0.05 mg/mL bovine pituitary extract and 5 ng/mL human recombinant epidermal growth factor. All cells were cultured in a humidified incubator at 37 °C with 5% $CO_2$. All cancer cell lines were obtained from the Cancer Cell Line Encyclopedia (CCLE) distributed by the Broad Institute Biological Samples Platform, except HK-2 and HEK-293T cells that were obtained from American Type Culture Collection (ATCC). All cells were regularly tested for mycoplasma contamination using MycoAlert Plus (Lonza) and cells used in experiments were negative for mycoplasma.

**Compound sources, synthesis, and treatment**. ML210 and RSL3 were synthesized according to previously described protocols[15,62]. The concentration range for ML210 was 0.01953–20 μM in 11-concentration experiments and 0.07813–5 μM in 7-concentration experiments unless otherwise indicated. The concentration range for RSL3 is 0.001953–2 μM in 11-concentration experiments and 0.01563–1 μM in 7-concentration experiments unless otherwise indicated. Liproxstatin-1 (Lip-1, Sigma-Aldrich SML1414) was used at 0.5–1 μM. Ferrostatin-1 (Fer-1, Sigma-Aldrich, SML0583) was used at 1–5 μM. Polyunsaturated acids (arachidonic acid, C20:4) (Cayman Chemicals) were conjugated with fatty-acid free BSA (Sigma-Aldrich) using previously described protocols[63], and were applied to cell culture media at 20 μM for 3 days.

**Gene-expression analysis by qRT-PCR**. Total RNA was extracted from cells using RNeasy Mini kit (Qiagen) following the manufacturer's instructions. cDNA was synthesized using the ProtoScript First Strand cDNA Synthesis kit (New England Biolabs). Quantitative PCR reaction mixtures were prepared with SYBR Green PCR Master Mix (Thermo Fisher Scientific, Applied Biosystems). PCR reactions were performed and analyzed on a LightCycler 480 Instrument (Roche).

Each sample condition contains at least three biological replicates and all measurements were performed with four technical replicates. Mean and standard deviation (s.d.) of biological replicates were presented unless otherwise indicated. Gene-expression levels were first normalized to internal control genes including *B2M* and *GAPDH* and then to no-perturbation conditions unless otherwise indicated. qPCR primers used in this study are listed in Supplementary Data 8.

**Immunoblotting and antibodies**. Adherent cells were briefly washed twice with PBS and lysed with 1% SDS lysis buffer containing 10 mM EDTA and 50 mM Tris-HCl, pH 8.0. Lysates were collected, briefly sonicated, and incubated at 95 °C for 10 min, and the protein concentrations were determined by BCA Protein Assay kit (Pierce). Calibrated samples were diluted with 4× LDS sampling buffer (Novus), separated by SDS-PAGE using NuPAGE 4–12% Bis-Tris protein gels (Novus), and transferred to nitrocellulose or PVDF membranes by iBlot2 protein-transfer system (Thermo Fisher Scientific). Membranes were blocked with 50% Odyssey blocking buffer (LiCor) diluted with 0.1% Tween-20-containing TBS and immunoblotted with antibodies from Abcam, including GPX4 (Ab41787), HIF-1α (Ab51608), HILPDA (Ab78349), ACSL4 (Ab155282), and from Cell Signaling Technologies, including V5-tag (D3H8Q, #13202), HIF-1β/ARNT (D28F3, #5537), HIF-2α (D9E3, #7096), KEAP1 (D6B12, #8047), NRF2 (D1Z9C, #12721), β-Actin (8H10D10, #3700 and 13E5, #4970). All antibodies were diluted at 1:1000 for immunoblotting. Membranes were then washed with TBST and incubated with IRDye 800CW goat-anti-Rabbit or 680RD donkey-anti-Mouse secondary antibodies (LiCor). Immunoblotting images were acquired on an Odyssey equipment (LiCor) according to the manufacturer's instructions, and analyzed in the ImageStudio software (LiCor). Unless otherwise indicated, β-Actin was used as a loading control. Raw, full scan images are presented in Supplementary Figs. 8 and 9.

**CRISPR/Cas9-mediated genome editing and RNA interference**. For CRISPR/Cas9-mediated genome-editing, cells were engineered for Cas9 expression with pLX-311-Cas9 vector (Addgene 96924), which contains the blasticidin S-resistance gene driven by the SV40 promoter and the SpCas9 gene driven by the EF1α promoter. sgRNA sequences were cloned into the pLV709 doxycycline-inducible or pXPR_BRD050 constitutive sgRNA expression vectors. For shRNA-mediated RNA interference, shRNAs targeting the genes of interest were pre-cloned into constitutive shRNA expression vectors pLKO.1 or pLKO-TRC005 by the Broad Institute Genetic Perturbation Platform. Lentiviruses were generated from sgRNA/shRNA constructs in HEK-293T packaging cells in 96-well plate format using FUGENE6 (Promega) as the transfection reagent and infected cells for sgRNA or shRNA expression. Infected cells were selected with puromycin at 2 μg/mL starting 48 h post-infection and propagated for further analysis. Cells transduced with inducible sgRNA constructs were treated with 1 μg/ml doxycycline (Sigma-Aldrich) for 7–14 days prior to gene-knockout validation by immunoblot analysis. Sequences for sgRNAs and shRNAs used are listed in Supplementary Data 9 and 10, respectively.

**Single-cell cloning**. 786-O-Cas9 cells were transfected with ribonucleoprotein (RNP) complex containing EnGen Cas9 NLS, *S. pyogenes* (New England Biolabs), Alt-R CRISPR tracrRNA and Alt-R CRISPR crRNA (Integrated DNA Technologies) according to the manufacturer's instructions using Lipofectamine RNAiMAX transfection reagent (Thermo Fisher Scientific). Transfected cells were sorted into 96-well plates at 1 cell/well on SONY SH800 cell sorter (SONY). Cells were allowed to grow for 7 days to become single-cell clones and analyzed by immunoblot for effective gene knockout and further analysis. The names of each clone were designated by the original plate numbers and well positions.

**Genome-wide CRISPR screen and data analysis**. Pooled lentiviruses for the sgRNA library was prepared using HEK-293T cells in T175 flasks as previously described[64]. Briefly, the viral titer and volume was pre-determined with pilot experiments prior to screening to ensure about 30% infection rate in the screening experiment and caution was taken to minimize multiple constructs transduced into the same cell. Optimized surface area for cell growth was pre-determined in pilot experiments to avoid reaching over-confluence. Puromycin concentration was pre-determined with pilot experiments before the screen. For the screening experiment, 150 million 786-O-Cas9 cells were infected with a lentiviral library containing 77,441 sgRNA targeting ~18,000 genes in the human genome to ensure each sgRNA is represented by at least 500 cells on average[65]. For the infection, cells added with the calculated lentivirus volume were supplemented with 4 μg/mL of polybrene and centrifuged at 930 RCF for 2 h. Fresh media was added at a 1:1 ratio post-centrifugation. Infected cells were selected with 2 μg/mL of puromycin for 96 h, expanded for another 4 days to reach the desired cell number and split for DMSO or 5 μM ML210 treatment. At least 40 million cells were used for each treatment condition to keep the minimum presentation number for each sgRNA above 500. Cells were exposed to ML210 treatment for 4, 6, or 8 days before being cultured in drug-free media for recovery and expansion for 24 h. Genomic DNA from cell pellets was purified using the QIAamp DNA Blood Maxi/Midi/Mini kits (Qiagen) according to the manufacturer's protocols and quantified using a Nanodrop 2000 (Thermo Fisher Scientific).

The sequencing library was prepared, sequenced, and analyzed as previously described[64,66]. Briefly, sgRNA cassettes were PCR-amplified and barcoded with sequencing adaptors utilizing ExTaq DNA Polymerase (Clontech). PCR products were purified with Agencourt AMPure XP SPRI beads (Beckman Coulter A63880) according to the manufacturer's instructions, quantified using a Nanodrop 2000, pooled into a master sequencing pool, and sequenced on a HiSeq sequencer (Illumina) with 300 nt single-end reads, loaded at 60% with a 5% spike-in of PhiX DNA.

For CRISPR screen data analysis, the sgRNA sequences were mapped to a reference file containing all SpCas9 sgRNAs in the library and the sgRNA-associated barcodes were counted and mapped to the barcode reference file. The read count matrix was normalized to reads per million reads (RPM) within each condition. Normalized read counts for each sgRNA in the ML210-treated conditions were compared with those in the DMSO-treated condition. SgRNA and gene level analyses for each condition are listed in Supplementary Data 1–3. Genes represented by 3–10 sgRNAs were included in the differential enrichment analysis. Genes of interest in each ML210-treated condition were required to have at least 2 sgRNAs exhibiting at least 2-fold enrichment with p-values <0.05. Selected genes that scored in all three ML210-treated conditions were further validated with shRNA-mediated knockdown.

**Lipid peroxidation analysis by imaging and flow cytometry.** For imaging, cells were plated at 5000 cells per well in a 96-well, Cell Carrier Ultra Microplate (PerkinElmer) in the appropriate cell culture media, supplemented with 1 µM of liproxstatin-1 where indicated and cultured overnight. Cells were incubated with 0.1% DMSO (same volume of DMSO as samples with a compound), 10 µM ML210, or 10 µM ML210 + 1 µM of liproxstatin-1 for the indicated times (1–4 h). During the last hour of incubation, the media also contained 60 nM DRAQ7 (Abcam), 1 µg/mL Hoechst 33342 (Thermo Fisher Scientific), and 1 µM BODIPY-581/591 C11 (Thermo Fisher Scientific) for live-cell imaging. Cells were imaged at ×63 magnification using an Opera Phenix High-Content Screening System (PerkinElmer, Waltham, MA) equipped with 405, 488, 560, and 647 nm lasers. Image analysis was conducted with Harmony software (PerkinElmer). All images (14 images per well) were collected with the same instrument parameters and processed with the same settings to maximize the ability to compare results between conditions. 9 randomly selected images per condition were presented in Supplementary Figure 1.

For flow cytometry analysis, 786-O, 769-P, and derivative cells were treated with DMSO or ML210 (5 µM) for 90 min, while for the last 45 min cells were also incubated with 5 µM of BODIPY-C11 dye. Before flow cytometry, cells were washed with PBS twice, stained with Hoechst 33342 for 5 min, trypsinized and filtered into single-cell suspensions. Flow cytometry analysis was performed on a SONY SH800 cell sorter with standard settings, using PE-TexasRed filter for reduced BODIPY-C11 (emission: 590 nm) and the FITC filter for oxidized BODIPY-C11 (emission: 510 nm). A minimum of 10,000 cells were analyzed for each condition, and each experiment was independently performed at least twice and representative experimental results are shown. Data analysis was performed using the FlowJo 10 software. An example gating strategy is demonstrated in Supplementary Fig. 4.

**CellTiter-Glo assay for viability analysis.** For cellular viability assays, cells were seeded in 384-well opaque white tissue culture and assay plates (Corning) at 1000 cells/well. 18–24 h after seeding, cells were treated with compounds at indicated concentrations for 48–72 h. Cellular ATP levels were quantified using CellTiter-Glo (Promega) on a multi-plate reader (Envision). Relative viability was normalized to the respective DMSO-treated condition unless otherwise indicated. For data presentation, the mean and standard deviation (s.d.) for the four biological replicates of each data point in a representative experiment is presented. Sigmoidal non-linear regression models were used to compute the regression fitting curves. For plate-based screening, area-under-curve (AUC) value for each regression curve is calculated and normalized to 1 as the total AUC for the concentration ranges of ML210 or RSL3.

**Lipid droplet abundance analysis by flow cytometry.** 786-O derivative cell lines were grown at optimal confluence and stained with BODIPY-493/503 (Thermo Fisher Scientific, Molecular Probes) at 2 µM final concentration for 30 min according to the manufacturer's instructions. Cells were then briefly washed with PBS, trypsinized, collected, stained with Hoechst 33258 for 5 min and filtered through a 70 µm nylon filter. The resulted post-staining single-cell suspensions were analyzed on a SONY SH800 cell sorter (SONY) according to the manufacturer's protocols. The filter used for detecting the BODIPY-493/503 signal was FITC 488 nm. A minimum of 10,000 cells were analyzed for each condition, and each experiment was independently performed at least twice and representative experimental results are shown. An example gating strategy is demonstrated in Supplementary Fig. 7.

**cDNA overexpression screening of HIF-2α-dependent genes.** cDNAs for the HIF-2α-dependent genes identified by RNA-Seq were obtained from the previously described human cDNA library collection at the Broad Institute Genetic

Perturbation Platform (Supplementary Data 6)[67]. These cDNAs were constructed for mammalian expression in the pLX-TRC317 vector system. Lentiviruses were produced with the cDNA constructs in HEK293-T cells in 96-well format. EPAS1$^{-/-}$ 786-O clones were infected with the cDNA lentivirus array, selected with 2 µg/mL puromycin for 96 h and analyzed for the cells' ferroptosis susceptibility 7 days post-infection. Control vectors expressing EGFP were used to evaluate the infection rate. Genes of interest were identified as having significantly shifted ML210 and RSL3 sensitivity curves compared with EGFP-expressing EPAS1$^{-/-}$ 786-O cells. Protein expression of the top hit genes and controls was verified by immunoblot analysis.

**RNA-Seq and data analysis.** RNA-Seq analysis was performed with wildtype and three EPAS1$^{-/-}$ 786-O single-cell clones generated by CRISPR/Cas9 to identify the HIF-2α-responsive genes. Total RNA was extracted from adherent 786-O cells and derivatives using the RNeasy Mini Kit (Qiagen) according to the manufacturer's instructions. The RNA sequencing library was prepared using NEB-Next Ultra RNA Library Prep Kit following the manufacturer's recommendations. Briefly, mRNAs were first enriched with Oligo-d(T) beads and fragmented for 15 min at 94 °C. First strand and second strand cDNA library was subsequently synthesized, end repaired, and adenylated at 3' ends. Universal adapter was ligated to cDNA fragments, followed by index addition and library enrichment with limited-cycle PCR. Sequencing libraries were validated using the Agilent Tapestation 4200, and quantified using a Qubit 2.0 Fluorimeter as well as by quantitative PCR. The sequencing libraries were multiplexed and clustered on one lane of a flowcell. After clustering, the flowcell was loaded on the Illumina HiSeq instrument according to the manufacturer's instructions. The samples were sequenced using a 2 × 150 bp paired-end configuration. Image analysis and base calling were conducted by the HiSeq Control Software. Raw sequence data generated from Illumina HiSeq was converted into fastq files and de-multiplexed using Illumina's bcl2fastq 2.17 software. One mismatch was allowed for index sequence identification.

For RNA-Seq data analysis, raw paired-end 150 bp/150 bp sequencing reads were mapped to human genome build hg19 using Bowtie2 (v2.3.1) with standard settings. On average 66% of read pairs were uniquely mapped to the hg19 genome. Mapped reads were counted to gene features by the htseq-count function from HTSeq (version 0.9.1) with standard settings, normalized to library size and analyzed for differentially expressed genes with DESeq2 (Bioconductor). Heatmaps were generated using the heatmap.2 function in gplots package in R (The Comprehensive R Archive Network).

**Lipidomic profiling and data analysis.** Analyses of polar and non-polar lipids were conducted using an LC-MS system comprising a Shimadzu Nexera X2 U-HPLC (Shimadzu Corp.) coupled to an Exactive Plus orbitrap mass spectrometer (Thermo Fisher Scientific). Lipids were extracted from cells with 0.8 mL iso-propanol (HPLC Grade; Honeywell). Three replicates were analyzed for each cell line or condition. Cell extracts were centrifuged at 10,000 RCF for 10 min to removed residual cellular debris prior to injecting 10 µL onto an ACQUITY BEH C8 column (100 × 2.1 mm, 1.7 µm; Waters). The column was eluted isocratically with 80% mobile phase A (95:5:0.1 vol/vol/vol 10 mM ammonium acetate/methanol/formic acid) for 1 min followed by a linear gradient to 80% mobile-phase B (99.9:0.1 vol/vol methanol/formic acid) over 2 min, a linear gradient to 100% mobile phase B over 7 min, then 3 min at 100% mobile-phase B. MS data were acquired using electrospray ionization in the positive-ion mode over 200–1100 $m/z$ and at 70,000 resolutions. Other MS settings were: sheath gas 50, in source CID 5 eV, sweep gas 5, spray voltage 3 kV, capillary temperature 300 °C, S-lens RF 60, heater temperature 300 °C, microscans 1, automatic gain control target 1e6, and maximum ion time 100 ms. Raw data were processed using TraceFinder 3.3 (Thermo Fisher Scientific) and Progenesis QI (Nonlinear Dynamics) software for detection and integration of LC-MS peaks. Lipid identities were determined based on comparison to reference standards and reference plasma extracts and were denoted by total number of carbons in the lipid acyl chain(s) and total number of double bonds in the lipid acyl chain(s).

For lipidomics data analysis, median normalization was performed between each sample in the same experiment. Median-normalized lipidomic datasets are presented in Supplementary Data 4, 5 and 7. Differential-abundance analysis was performed between previously annotated lipid species (about 200 lipids were previously annotated) using two-tailed Student's T-test. For fold-change analysis, each dataset was normalized to the mean of the WT cell condition for each lipid species, and the ratio between Test/WT was log2 transformed and presented as heatmaps, bar graphs or volcano plots. P values are adjusted for multiple-test correction using Benjamini-Hochberg correction method and presented as -log10 adj. p. For principal component plots, the log-ratios between Test/WT for each lipid were further median normalized and computed using the principal component analysis function in DESeq2 (Bioconductor) using RStudio.

**Public dataset queries.** The Cancer Therapeutics Response Portal (portals.broadinstitute.org/ctrp/) compound sensitivity dataset, a data matrix containing the normalized AUC values of each compound in each cell line, was downloaded from [https://ocg.cancer.gov/programs/ctd2/data-portal], (also deposited in [ftp://caftpd.nci.nih.gov/pub/OCG-DCC/CTD2/Broad/] and [https://github.com/

remontoire-pac/ctrp-reference/tree/master/auc]). For the lineage/histotype-specific analyses, compounds that were profiled in at least 2/3 of the total solid cancer cell line collection, and cell lines that were profiled using at least 50% of the compounds, were included in the statistical analyses. Primary cancer types that contain >5 cell lines profiled for compound sensitivities were presented. Mann–Whitney–Wilcoxon tests were performed between cancer cell lines from each cancer type and other solid cancer cell line collections. Statistical significances were adjusted for multiple-test correction using the Benjamini-Hochberg correction method. Compound sensitivity-gene expression correlation analysis were performed with the web tools in CTRP [portals.broadinstitute.org/ctrp/][11].

For gene expression analysis of CTRP cell lines, RNA-Seq dataset was downloaded from the Cancer Cell Line Encyclopedia data portal [https://portals.broadinstitute.org/ccle/data]. For Cancer Dependency Map (DepMap) datasets, dependency scores for each gene of interest, including CERES scores from CRISPR/Cas9 screening experiments and DEMETER scores for RNA interference screening experiments, were downloaded from the DepMap web portal [https://depmap.org/portal]. ChIP-Seq dataset for HIF-2α/HIF-1β in 786-O cells (GSE34871) was downloaded from Gene Expression Omnibus [https://www.ncbi.nlm.nih.gov/geo/query/acc.cgi?acc=GSE34871].

**Animal studies**. All animal experiments were in compliance with relevant ethical regulations and were approved by the Institutional Animal Care and Use Committee (IACUC) at the Broad Institute. Briefly, 3–4-week-old, male athymic nude mice were used for 786-O xenograft experiments. Five million cells for each injection were trypsinized, resuspended in 50 μl PBS containing 0.5 μM DMSO or Lip-1, mixed with 50 μl Matrigel (BD Biosciences), and implanted to mice subcutaneously. For Lip-1 treatment, Lip-1 was first dissolved in DMSO then diluted with PBS and injected into mice at 20 mg/kg body weight daily. Lip-1 or vehicle treatment was continued for 10 days before withdrawal. Tumor sizes were monitored and measured on a weekly basis. Tumor volumes were quantified by measuring the length (L) and width (W) of the tumor using a caliper and calculated according to $V = (L*W*W)/2$.

**Patient-derived cancer cell model generation**. Human renal cell carcinoma samples, CCLF_KIPA_0001_T and CCLF_KIPA_0002_T, were obtained from patients, with their informed consent, at the Dana Farber Cancer Institute (DFCI); and all procedures were conducted under an Institutional Review Board (IRB)-approved protocol. Clinical information for the patient tumors were included in Supplementary Fig. 1. For primary cell line generation, tumor resections were placed in a sterile conical tube containing DMEM media (Thermo Fisher Scientific, cat. #11995073) with 10% FBS (Sigma Aldrich, cat. #F8317), 1% penicillin–streptomycin (Thermo Fisher Scientific, cat. #15140163), 10 μg/ml of gentamicin, and 250 ng/ml fungizone on wet ice during transport from the operating room to the research laboratory. Resections were placed in a 15 ml conical flask with 5 ml DMEM media, 10% FBS, 1% penicillin–streptomycin, and the digestion enzymes regular collagenase 1 ml (StemCell, cat. #07912) and dispase 1 ml (StemCell Technologies, cat. #07913). The flask was placed on a rotator and incubated at 37 °C for 1 h. The cells were then centrifuged at 200 × g (RCF) for 5 min. Cell pellets were resuspended in a 50:50 mix culture medium of Smooth Muscle Growth Medium-2 (Lonza CC-3182) and ACL4 media with 5% FBS[68,69]. Later, suspended cells were plated into a 96-well plate. The medium was changed every 3 days, and cells were maintained at 37 °C in a humidified 5% CO2 incubator. RCC cells were passaged using Gibco TrypLE Express (Thermo Fisher Scientific, cat. # 12604039) to detach cells when the cells reached 80–90% confluence. CCLF_KIPA_0001_N cells were prepared with a similar protocol but cultured with conditional media as previously described[70].

**Statistical analysis**. Data are generally expressed as mean ± s.d. unless otherwise indicated. No statistical methods were used to predetermine sample sizes. Statistical significance was determined using a two-tailed Mann–Whitney–Wilcoxon test or two-tailed Student's T-test using Prism 7 software (GraphPad Software) unless otherwise indicated. The Benjamini-Hochberg correction method was used to adjust the p-values where multi-testing corrections were involved. Statistical significance was set at $p < 0.05$ unless otherwise indicated.

**Reporting summary**. Further information on experimental design is available in the Nature Research Reporting Summary linked to this article.

## Data availability

Raw sequencing files and read count matrix for the CRISPR screening experiment (related to Fig. 2) is deposited in Gene Expression Omnibus under the accession number GSE126696, with the processed data matrix supplied in Supplementary Data tables. The accession number for raw RNA sequencing data (related to Fig. 4) in GEO is GSE115389. Original lipidomic profiling data are available in Supplementary Data tables (related to Figs. 3 and 4). Links to publicly available datasets are provided in the Public dataset queries section of Methods, with data analysis procedures described. All remaining data and computational code that support the findings of this study are available from the corresponding author (S.L.S.) upon request.

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

## Acknowledgements

We thank Tanaz Sharifnia, Shubhroz Gill, Olivia Bare, Xi Wang, Xin Jin, Xin Rong, and Robert L. Bowman for discussion, and members of Project Achilles for sharing Cas9-expressing cell lines. We thank C. Suk-Yee Hon for project management and organization assistance. We thank the Eric Jonasch laboratory for the HIF-2α-GFP plasmid. We thank John A. Steinharter for coordinating the patient sample collection. We acknowledge funding support for M.J.P. from the NIH NHLBI award T32HL007627, for V.S.V. from the NCI SPORE P50 CA101942-13 Directors Choice Award. This work was supported by the NCI's Cancer Target Discovery and Development (CTD$^2$) Network (grant number U01CA217848, awarded to S.L.S.), and in part by NIH/NCI DF/HCC Kidney Cancer SPORE P50 CA101942 to S.S. and T.K.C., and by the Trust Family, the Kohlberg chair at Harvard Medical School, the Loker Pinard and the Michael Brigham Funds at the Dana-Farber Cancer Institute for Kidney Cancer Research to T.K.C.

## Author contributions

Y.Z. and S.L.S. conceived the project, designed the experiments and wrote the manuscript. Y.Z. performed experiments and bioinformatics analyses. A.A.D. and C.B.C. assisted the metabolomics analyses. J.D. assisted the genetic screening experiments and data analysis. M.J.P. and M.K-A. assisted the imaging experiments. H.L., V.D., E.S.L., V.S.V., B.K.W., and P.A.C. assisted in bioinformatics analysis and manuscript preparation. J.K.E. and W.W. performed chemical synthesis. Y.-Y.T., R.D., S.S., T.K.C., and J.S.B. established the patient-derived cell line models. All authors interpreted data, discussed results and contributed to writing the manuscript.

## Additional information

**Competing interests:** S.L.S. is a member of the Board of Directors of the Genomics Institute of the Novartis Research Foundation (GNF); a shareholder and member of the Board of Directors of Jnana Therapeutics; a shareholder of Forma Therapeutics; a shareholder of and adviser to Decibel Therapeutics and Eikonizo Therapeutics; an adviser to Eisai, Inc., the Ono Pharma Foundation, and F-Prime Capital Partners; and a Novartis Faculty Scholar. P.A.C. is an adviser to Pfizer, Inc. S. Signoretti has consulting or advisory role for AstraZeneca/MedImmune, Merck, AACR, NCI; royalties from Biogenex Laboratories; and research funding from AstraZeneca, Exelixis, Bristol-Myers Squibb. T.K.C. receives institutional and personal research funds from AstraZeneca, Bayer, BMS, Cerulean, Eisai, Foundation Medicine Inc., Exelixis, Ipsen, Tracon, Genentech, Roche, Roche Products Limited, GlaxoSmithKline, Merck, Novartis, Peloton, Pfizer, Prometheus Labs, Corvus, Calithera, Analysis Group, Takeda; and receives personal honoraria from AstraZeneca, Alexion, Sanofi/Aventis, Bayer, BMS, Cerulean, Eisai, Foundation Medicine Inc., Exelixis, Genentech, Roche, GlaxoSmithKline, Merck, Novartis, Peloton, Pfizer, EMD Serono, Prometheus Labs, Corvus, Ipsen, Up-to-Date, NCCN, Analysis Group,

NCCN, Michael J. Hennessy (MJH) Associates, Inc (Healthcare Communications Company with several brands such as OnClive and PER), L-path, Kidney Cancer Journal, Clinical Care Options, Platform Q, Navinata Healthcare, Harborside Press, American Society of Medical Oncology, NEJM, Lancet Oncology, Heron Therapeutics; and has consulting or advisory role for AstraZeneca, Alexion, Sanofi/Aventis, Bayer, BMS, Cerulean, Eisai, Foundation Medicine Inc., Exelixis, Genentech, Heron Therapeutics, Roche, GlaxoSmithKline, Merck, Novartis, Peloton, Pfizer, EMD Serono, Prometheus Labs, Corvus, Ipsen, Up-to-Date, NCCN, Analysis Group. No speaker's bureau. No leadership or employment in for-profit companies. Other present or past leadership roles for T.K.C.: Director of GU Oncology Division at Dana-Farber and past President of medical Staff at Dana-Farber), member of NCCN Kidney panel and the GU Steering Committee, past chairman of the Kidney cancer Association Medical and Scientific Steering Committee). No Patents, royalties or other intellectual properties. Travel, accommodations, expenses, in relation to consulting, advisory roles, or honoraria. Medical writing and editorial assistance support may have been funded by Communications companies funded by pharmaceutical companies. The institution (Dana-Farber Cancer Institute) may have received additional independent funding of drug companies or/and royalties potentially involved in research around the subject matter. CV provided upon request for scope of clinical practice and research. The remaining authors declare no competing interests.

