## [Peer Review File · Nature Communications]

REVIEWERS' COMMENTS:

Reviewer #1 (Remarks to the Author):

The authors have responded adequately and I have no further concerns.

Reviewer #2 (Remarks to the Author):

The authors have appropriately addressed all my concerns and have discussed their findings in light of recent papers on the role of the HIF system in the context of ferroptosis. Therefore, I have no further comments.

Reviewer #3 (Remarks to the Author):

The authors have thoroughly responded to my concerns and the paper is now an excellent fit for this journal.

Reply to referee comments:

Response: We thank all the reviewers for their time and we are delighted that all of you are satisfied with our revisions.

REVIEWERS' COMMENTS:

Reviewer #1 (Remarks to the Author):

The authors have responded adequately and I have no further concerns.

Reviewer #2 (Remarks to the Author):

The authors have appropriately addressed all my concerns and have discussed their findings in light of recent papers on the role of the HIF system in the context of ferroptosis. Therefore, I have no further comments.

Reviewer #3 (Remarks to the Author):

The authors have thoroughly responded to my concerns and the paper is now an excellent fit for this journal.